# Alterations in Gut Microbiota After Upper Gastrointestinal Resections: Should We Implement Screening to Prevent Complications?

**DOI:** 10.3390/medicina61101822

**Published:** 2025-10-11

**Authors:** Urška Novljan, Žan Bohinc, Niko Kaliterna, Uroš Godnov, Tadeja Pintar Kaliterna

**Affiliations:** 1General Hospital Novo Mesto, 8000 Novo Mesto, Slovenia; ursanovljan@gmail.com; 2Medical Faculty, University of Ljubljana, 1000 Ljubljana, Slovenia; bohinc.zan@gmail.com; 3Medical Faculty, University of Maribor, 2000 Maribor, Slovenia; niko.kaliterna@gmail.com; 4Department of Computer Science, Faculty of Mathematics, Natural Sciences and Information Technologies, University of Primorska, 6000 Koper, Slovenia; uros.godnov@famnit.upr.si; 5Department of Abdominal Surgery, Division of Surgery, University Medical Centre Ljubljana, 1000 Ljubljana, Slovenia

**Keywords:** small intestinal bacterial overgrowth (SIBO), gut dysbiosis, intestinal microbiota, upper GI surgery, glucose–hydrogen breath test (GHBT), exocrine pancreatic insufficiency (EPI), pancreatic cancer, gastric cancer, Crohn’s disease, metabolic bariatric surgery, postoperative complications

## Abstract

*Background*: Surgical procedures and alterations of the gastrointestinal (GI) tract increase the risk of small intestinal bacterial overgrowth (SIBO), which is associated with GI symptoms and complications that compromise postoperative recovery. However, the prevalence and clinical impact of SIBO after various upper GI surgical procedures remain poorly understood. *Objective*: This study aimed to evaluate the prevalence of SIBO after different types of upper GI surgery and to investigate the associated clinical factors. *Methods*: We conducted an observational study involving 157 patients with a history of upper GI surgery: Roux-en-Y gastric bypass (RYGB), laparoscopic single-anastomosis gastric bypass (OAGB), subtotal (STG) or total gastrectomy (TG), subtotal (SP)or total pancreatectomy (TP), cephalic duodenopancreatectomy (WR), and small bowel resection for Crohn’s disease. A glucose–hydrogen breath test was performed, and demographic, clinical, and treatment-related data were collected. Statistical analyses included *t*-tests, non-parametric tests, ANOVA, and correlation analyses using R software. *Results*: At a median follow-up of 25.7 ± 18.1 months, 31% (48/157) of patients tested positive for SIBO. The highest prevalence was observed after RYGB and OAGB (43%), followed by TG (30%), STG (29%), TP/WR (28%), and Crohn’s disease bowel resection (19%). No cases of SIBO were observed after SP. SIBO positivity was significantly associated with bloating and flatulence (*p* = 0.002), lactose intolerance (*p* = 0.047), systemic sclerosis (*p* = 0.042), T2D (*p* = 0.002), and exposure to adjuvant chemotherapy (*p* = 0.001) and radiotherapy (*p* = 0.027). In addition, the risk of SIBO increased proportionally with the duration of GI resection or exclusion (*p* = 0.013). *Conclusions*: In our study, the prevalence of SIBO after upper GI surgery was 31%, with the highest incidence (43%) observed in metabolic surgery patients. Importantly, adjuvant radio/chemotherapy was associated with an increased risk of SIBO, and extensive small bowel resection or exclusion was strongly associated with an increased risk of SIBO. Furthermore, the limitations of current diagnostic methods, which lack sufficient sensitivity and specificity, highlight the importance of early screening and standardization of diagnostic techniques to improve patient management and outcomes.

## 1. Introduction

The human gut microbiota is a complex and dynamic ecosystem comprising bacteria, archaea, viruses, and unicellular eukaryotes, with bacteria accounting for about 90% of the microbial population [1,2,3]. This intricate microbial network is crucial for maintaining gastrointestinal (GI) and systemic homeostasis through metabolic, immunological, and neuromodulator functions. However, upper GI surgery and metabolic interventions can significantly alter the gut microbiome and lead to gut dysbiosis, which disrupts its composition and function [4,5,6].

Small intestinal bacterial overgrowth (SIBO) is a clinical disorder and a type of dysbiosis characterized by an overgrowth of colonic bacteria in the small intestine [7,8,9,10,11]. SIBO can be asymptomatic, although non-specific GI symptoms such as abdominal pain, bloating, flatulence, diarrhea, constipation, malabsorption, and their sequelae are common [4,12]. The clinical symptoms often overlap with other conditions such as exocrine pancreatic insufficiency (EPI), type 2 diabetes mellitus (T2DM), and certain eating disorders (anorexia nervosa, bulimia), which are associated with an increased incidence of SIBO.

The current gold standard for diagnosing SIBO is a jejunal aspirate culture showing ≥10^5^ colony-forming units (CFU)/mL of colonic-type bacteria. However, its invasiveness, technical complexity, and susceptibility to contamination limit its clinical applicability. Non-invasive breath tests (BT), particularly hydrogen/methane BT, offer a more viable alternative, although their diagnostic accuracy is compromised by a lack of standardization, particularly regarding substrate selection, dosing, and interpretation thresholds [13].

Physiological mechanisms such as gastric acid secretion, bile flow, intestinal motility, mucus production, and mucosal immunity collectively limit bacterial colonization of the small intestine. The ileocecal valve serves as an additional anatomical barrier, preventing retrograde bacterial translocation from the colon [14]. Surgical alterations of the upper GI tract impair these defense mechanisms, especially in the context of blind loops (e.g., after Billroth II reconstruction or metabolic surgery with long bypass procedures), intestinal stasis, or bypass segments, while metabolic and immunological changes further exacerbate microbial dysregulation [14,15,16]. Diseases such as diabetes mellitus, scleroderma, inflammatory bowel disease (IBD), Helicobacter pylori infections, and the use of proton pump inhibitors further increase the risk of SIBO [15,16].

Given the high prevalence and clinical impact of SIBO in surgical patients, early detection and intervention are of paramount importance. The aim of this study was to determine the prevalence of SIBO in a well-defined cohort of patients who had undergone upper GI tract surgery for metabolic bariatric indications, gastric cancer, pancreatic cancer, or Crohn’s disease (CD). In addition, we aimed to identify the association between SIBO and postoperative GI symptoms and to highlight the need for standardized diagnostic approaches and targeted therapeutic strategies to optimize postoperative outcomes and long-term quality of life.

## 2. Methods

### 2.1. Study Design

A prospective observational study was conducted with patients identified through a pooled surgical database (Birpis, Naklo, Slovenia), which included all individuals who had undergone predefined upper GI procedures between January 2017 and June 2022. The following procedures were considered: Roux-en-Y gastric bypass (RYGBP), laparoscopic one-anastomosis gastric bypass (OAGB), subtotal gastrectomy (STG), total gastrectomy (TG), subtotal pancreatectomy (SP), total pancreatectomy (TP), cephalic duodenopancreatectomy (WR), and small bowel resection for CD.

From this database, patients were stratified by surgical subgroup to ensure adequate representation, and then randomly invited to participate in the study via telephone call or e-mail. Between January 2021 and June 2022, a total of 157 patients underwent a standardised glucose–hydrogen BT and completed structured questionnaires on demographics, clinical status, and GI symptoms.

Although the database was used to identify eligible patients and ensure balanced subgroup sizes, all clinical and diagnostic data were collected prospectively according to a prespecified protocol.

### 2.2. Participants

Inclusion criteria were adults (≥18 years old) of both sexes, regardless of the presence or absence of GI symptoms. The number of participants in each surgical subgroup is shown in Table 1.

Exclusion criteria were inability to comply with study protocols, inability to complete the study (due to unconsciousness), recent use of antibiotics, prokinetics, or laxatives within two weeks prior to testing, and a basal hydrogen concentration greater than 10 ppm on two separate measurements 20 min apart. The flow chart for the participants is shown in Figure 1.

### 2.3. Glucose–Hydrogen Breath Test

SIBO was diagnosed using a standardised glucose–hydrogen (H_2_) BT with the Lactofan 2 Fischer^®^ device (Leipzig, Germany). Participants were instructed to follow a standardised low-fermentation diet for 24 h before the test and to refrain from smoking or exercising on the day of the test. After an overnight fast of 12 h, an exhaled air sample was collected, and results were expressed in parts per million (ppm). If the baseline H_2_ concentration was less than 10 ppm, participants ingested 25 g of glucose dissolved in 200 mL of water. Exhaled air samples were then collected every 20 min for 120 min (six measurements in total). An increase of 12 ppm or more in hydrogen concentration from baseline within 120 min was considered diagnostic for SIBO.

### 2.4. Questionnaire-Based Assessement

All participants completed a structured questionnaire consisting of four parts (see Appendix A for Questionnaire):Demographic and anthropometric data (custom-designed section);GI Symptom Rating Scale (GSRS)—adapted from the validated instrument by Dimenäs et al. [17];Small Intestinal Bacterial Overgrowth (SIBO) questionnaire—originally developed by Everyday Wellness Clinic. SIBO-related symptoms were assessed using this questionnaire, which has been in use since 2014 and includes items on GI symptoms, past treatments, diet and lifestyle factors, and comorbidities. The full questionnaire (with permission) is provided in Appendix A—Questionnaire SIBO (Everyday Wellness Clinic);36-Item Short Form Health Survey (SF-36)—a widely used and validated quality-of-life instrument.

The questionnaire was completed under the supervision of trained research staff. Data from the questionnaires were independently entered and coded by two researchers; any discrepancies were resolved by a third senior investigator.

### 2.5. Surgical Procedure

All surgical procedures were performed by experienced surgeons using standard techniques. The metabolic bariatric surgery was either RYGB or OAGB. In RYGB, a 50 cm biliopancreatic limb and a 100 cm jejunal exclusion (alimentary limb) were created with a linear and hand-sewn anastomosis. In OAGB, a longer gastric pouch of 15 cm, a 38F calibration tube, and a 150 cm jejunal exclusion were created. The gastrojejunal anastomosis corresponds to the diameter of the calibration tube in both surgical techniques. TG was performed with a Roux-en-Y reconstruction, as was STG with a standard jejunal exclusion (45–60 cm). SP included an en bloc resection of the pancreas, duodenum, common bile duct, and gallbladder, followed by pancreaticojejunostomy, hepaticojejunostomy, and gastrojejunostomy. TP was performed with a hepaticojejunostomy and gastrojejunostomy. Resection of the small bowel due to CD was performed with a segmental resection of the affected bowel and a hand-sewn or stapled anastomosis, using organ-preserving principles to minimise the risk of short bowel syndrome.

### 2.6. Statistical Analysis

Data were collected using Microsoft Excel (Microsoft Corp., Redmond, WA, USA) and analysed using R software (version 4.1.1) with the tidyverse and arsenal packages. Continuous variables were summarized as mean ± standard deviation (SD) or median with interquartile range (IQR), depending on the data distribution. Categorical variables were expressed as counts and percentages.

The Shapiro–Wilk test was used to assess the normality of data distribution. For group comparisons, Student’s *t*-test or Mann–Whitney U test (in the case of insufficient sample size) was applied for continuous variables, and the Chi-square or Fisher’s exact test for categorical variables, as appropriate.

Correlations were analyzed using Pearson’s correlation coefficient (r) for normally distributed variables and Spearman’s rank correlation coefficient (ρ) for non-normally distributed variables. Correlation strength was interpreted according to the following thresholds: 0.00–0.19, very weak; 0.20–0.39, weak; 0.40–0.59, moderate; 0.60–0.79, strong; 0.80–1.00, very strong correlation.

A *p*-value < 0.05 was considered statistically significant, with 95% confidence intervals calculated for all estimates.

## 3. Results

### 3.1. Participant Characteristics

A total of 157 patients underwent glucose–hydrogen BT between June 2021 and July 2022 following upper gastrointestinal surgery. The cohort included 56 patients after RYGB or LOAGB, 37 after STG or TG, 38 after SP or WR/TP, and 26 after small bowel resection for CD. The mean age of the study population was 56.5 ± 12.5 years (range 28–82), mean body weight was 78.0 ± 17.7 kg (range 50–147), and mean body mass index (BMI) was 26.6 ± 5.8 kg/m^2^.

### 3.2. Demographic Characteristics

No significant differences were observed between SIBO-positive (*n* = 48) and SIBO-negative (*n* = 109) patients in age (55.5 ± 12.9 vs. 56.9 ± 12.3 years, *p* = 0.535), sex (64.6% vs. 52.3% female, *p* = 0.153), BMI (27.2 ± 6.1 vs. 26.4 ± 5.7 kg/m^2^, *p* = 0.436), education level (4.52 ± 1.46 vs. 4.48 ± 1.30, *p* = 0.851), socioeconomic status (5.91 ± 1.71 vs. 5.97 ± 1.70, *p* = 0.848), or time from surgery to GHBT (26.1 ± 20.7 vs. 25.5 ± 16.9 months, *p* = 0.870) (Table 2). Subgroup analysis by surgical procedure confirmed the absence of significant demographic differences across the RYGB/LOAGB, STG/TG, SP/TP/WR, and CD resection groups (Table 2).

### 3.3. Prevalence of SIBO

Of the 157 patients, 48 (30.6%) were SIBO-positive on GHBT, while 109 (69.4%) were SIBO-negative (Figure 2). The prevalence of SIBO was highest after RYGB/LOAGB, with 24/56 patients (42.9%) testing positive. SIBO was detected in 2/7 patients (28.6%) after STG, 9/30 (30.0%) after TG, 0/9 (0.0%) after SP, 8/29 (27.6%) after TP/WR, and 5/26 (19.2%) after CD resection. The distribution of SIBO-positive and SIBO-negative cases across surgical subgroups is shown in Figure 3.

### 3.4. Symptoms

Compared with SIBO-patients (*n* = 109), SIBO+ patients (*n* = 48) reported significantly higher scores for bloating and flatulence (5.85 ± 3.08 vs. 4.28 ± 2.73, *p* = 0.002) and for bloating alone (2.87 ± 1.23 vs. 2.50 ± 1.05, *p* = 0.050) (Table 3).

No significant differences were observed for chronic pain (2.38 ± 1.91 vs. 2.38 ± 1.99, *p* = 0.997), diarrhea (3.19 ± 2.82 vs. 3.03 ± 2.51, *p* = 0.724), frequency of defecation (2.17 ± 1.00 vs. 2.04 ± 0.91, *p* = 0.428), obstipation (2.08 ± 2.22 vs. 1.82 ± 1.76, *p* = 0.422), abdominal cramps (2.64 ± 1.99 vs. 2.54 ± 2.09, *p* = 0.792), nausea (1.63 ± 1.35 vs. 1.79 ± 1.71, *p* = 0.557), vomiting (1.23 ± 0.78 vs. 1.34 ± 1.17, *p* = 0.552), reflux (2.71 ± 2.38 vs. 2.70 ± 2.45, *p* = 0.979), loss of appetite (1.56 ± 1.44 vs. 1.61 ± 1.61, *p* = 0.847), fatigue (3.74 ± 2.75 vs. 3.51 ± 2.73, *p* = 0.634), or other reported symptoms (*p* > 0.05 for all).

Symptom distribution by surgical subgroup is provided in Appendix A.

### 3.5. SIBO and Comorbidities

SIBO+ patients had significantly higher rates of lactose intolerance (0.688 ± 1.36 vs. 0.312 ± 0.940, *p* = 0.047), systemic sclerosis (0.06 ± 0.32 vs. 0.00 ± 0.00, *p* = 0.042), IBS (0.75 ± 1.30 vs. 0.32 ± 0.89, *p* = 0.018), and type 1/2 diabetes (1.13 ± 1.71 vs. 0.40 ± 1.15, *p* = 0.002) compared with SIBO-patients (Table 4). No significant differences were observed for joint pain, skin problems, rosacea, breathing problems, headache, or memory impairment (*p* > 0.05 for all).

### 3.6. SIBO and (Neo)Adjuvant Therapy

Adjuvant therapy was significantly associated with SIBO positivity (Table 5). SIBO+ patients had greater exposure to adjuvant chemotherapy (1.21 ± 0.419 vs. 1.64 ± 0.484, *p* = 0.001) and adjuvant radiotherapy (1.74 ± 0.452 vs. 1.93 ± 0.260, *p* = 0.027) compared with SIBO-patients.

No significant differences were observed for neoadjuvant chemotherapy (1.68 ± 0.478 vs. 1.79 ± 0.414, *p* = 0.374) or neoadjuvant radiotherapy (1.95 ± 0.229 vs. 1.95 ± 0.227, *p* = 0.988).

### 3.7. Length of Resection

Among 26 patients who underwent surgery for CD, ileocecal resection was performed in 15/26 (57.5%), ileum and terminal ileum resection in 1/26 (3.8%), segmental ileum resection in 2/26 (7.5%), small bowel resection in 3/26 (11.6%), segmental small bowel resection in 3/26 (11.6%), and total colectomy in 1/26 (3.9%). Data were not available for 1/26 (3.9%) patients.

Information on the exact length of bowel resection was obtained from surgical records in 20/26 (77%) patients. The risk of SIBO increased proportionally with the length of GI resection or exclusion (*p* = 0.013) (Figure 4).

## 4. Discussion

The gut microbiota forms a highly individualized ecosystem, and its disruption has been implicated in multiple GI and systemic disorders [5,18]. SIBO is a distinct form of gut dysbiosis, characterized by excessive bacterial colonization of the small intestine. It has been associated with GI symptoms, postoperative complications, and adverse nutritional and metabolic outcomes [6]. Surgical removal or bypass of GI segments alters luminal flow, motility, and antimicrobial defences, thereby increasing susceptibility to SIBO [5,9,10,19].

This study evaluated SIBO prevalence after extensive upper GI resections or modifications (RYGBP, LOAGBP, STG, TG, SP, WR, TP, and intestinal resections for CD), analysing symptoms, comorbidities, adjuvant treatments, and the influence of resection length. We found an overall SIBO prevalence of 31%, highest after bariatric surgery (43%), followed by gastric cancer resections (29–30%), pancreatic resections (28%), and CD resections (19%). To our knowledge, this is the first study to comprehensively evaluate SIBO after major upper GI surgeries. Comparisons with previous studies are difficult due to differences in surgical indications, patient selection, and diagnostic protocols [19]. Given this complexity, we structured the discussion into five sections: GI symptoms and comorbidities, bariatric surgery, gastric cancer resections, pancreatic surgery, and CD resections.

### 4.1. Symptoms and Comorbidities

Non-specific GI symptoms such as bloating, flatulence, abdominal discomfort, and nausea frequently complicate the postoperative course after upper GI surgery, with SIBO proposed as a potential cause due to excessive bacterial fermentation and gas production [20,21]. In our study, however, only bloating and flatulence occurred significantly more often in SIBO-positive patients, underscoring the limited diagnostic accuracy of symptoms alone. Beyond clinical presentation, several comorbidities—including altered GI anatomy (e.g., resections, strictures), impaired antimicrobial defenses (e.g., exocrine pancreatic insufficiency, achlorhydria), and delayed intestinal transit (e.g., diabetes mellitus, systemic sclerosis)—have been linked to increased SIBO risk [4]. Consistent with previous studies [22,23], we found significant associations between SIBO and systemic sclerosis (*p* = 0.042), diabetes mellitus (*p* = 0.002), irritable bowel syndrome (*p* = 0.018), and lactose intolerance (*p* = 0.047), highlighting the multifactorial nature of SIBO pathogenesis in postoperative patients.

### 4.2. Bariatric Surgery

RYGBP and LOAGBP exclude the proximal small intestine, create a blind loop, and delay bile exposure, thereby reducing the antimicrobial effects of bile acids, all of which are risk factors for SIBO [10,24]. The risk is already increased in obese patients even before surgery, with SIBO prevalence reported at 42% [25]. Our study showed a 43% SIBO prevalence after RYGBP and LOAGBP regardless of symptoms, similar to previous reports (~40%) [26], though some studies have reported higher rates (58–83%) in symptomatic cohorts [6,27]. Variations likely reflect differences in BT protocols, particularly glucose dose (25–75 g).

SIBO is increasingly recognised as a source of metabolic endotoxaemia, as increased intestinal permeability facilitates the translocation of bacterial products such as lipopolysaccharides (LPS) into the systemic circulation [28]. The liver, receiving portal blood directly from the gut, is therefore the first organ exposed to these microbial components. LPS-mediated activation of Toll-like receptor 4 (TLR-4) triggers NF-κB–dependent inflammatory cascades, leading to increased production of proinflammatory cytokines, including TNF-α, IL-1β, and IL-6 [28,29,30]. These mechanisms contribute not only to systemic inflammation and obesity-related complications but also to hepatic injury, promoting steatosis, ballooning degeneration, lobular and portal inflammation, fibrosis, and even progression towards metabolic-associated fatty liver disease (MAFLD) and non-alcoholic steatohepatitis (NASH) [28,31,32,33].

Furthermore, gut microbiota alterations may influence the gut–brain axis. Bacterial metabolites (e.g., SCFA) cross the blood–brain barrier, act on GPR41/GPR43 receptors, stimulate anorexigenic hormones GLP-1 and PYY, and modulate appetite regulation Via vagal pathways [34,35]. SIBO-related dysbiosis disrupts these mechanisms, reducing anorexigenic signalling and favouring hyperphagia and weight regain [34,36]. Given these findings, SIBO may underlie suboptimal long-term outcomes after bariatric surgery, including weight regain, type 2 diabetes recurrence, and rare but severe complications such as liver failure after long-limb bypass (>150 cm) [10,18]. Early SIBO detection and treatment, potentially including faecal microbiota transplantation, may improve metabolic outcomes [37].

### 4.3. Gastric Cancer Resection

Modification of normal GI anatomy after STG and TG creates a blind intestinal loop and disrupts gastric acid secretion, both of which predispose to SIBO [21,38,39]. In our study, SIBO prevalence reached 29–30% after STG and TG, whereas previous reports showed considerably higher rates (61.6–96.2%), particularly in symptomatic patients [21,38], likely reflecting methodological differences. We also found a higher prevalence of SIBO in patients receiving adjuvant chemotherapy or radiotherapy after gastric or pancreatic cancer resections. These treatments further compromise mucosal integrity by inducing ischaemic hypoxia, oxidative stress Via xanthine oxidase activation, and radiotherapy-mediated enterocyte necrosis, all of which impair epithelial barrier function and antimicrobial defences, thereby facilitating bacterial overgrowth [40,41]. Given that SIBO represents a potentially treatable cause of malabsorption, systematic testing after gastrectomy—especially in patients with weight loss, vitamin deficiencies, or persistent GI symptoms—appears clinically relevant.

### 4.4. Pancreatic Cancer Resection

Extensive pancreatic resections, including WR and TP, profoundly alter GI anatomy and reduce the secretion of antimicrobial peptides (AMPs), predisposing patients to SIBO. In our study, SIBO prevalence reached 28% after WR or TP, whereas no cases were observed after limited resections, suggesting that the extent of resection and preservation of pancreatic function strongly influence SIBO risk. Previous studies have reported altered gut microbiota composition after pancreatic surgery, including Klebsiella overgrowth and depletion of beneficial anaerobic species such as Ruminococcus, which are associated with postoperative complications such as fistulas [42,43,44].

AMPs play a key role in maintaining intestinal homeostasis, and their reduced secretion after major pancreatic surgery may further impair gut barrier function, favouring bacterial overgrowth and systemic inflammation [45]. When combined with SIBO, exocrine pancreatic insufficiency (EPI) creates a vicious cycle: SIBO-induced inflammation worsens pancreatic dysfunction, increases endotoxin exposure, and further impairs nutrient absorption, ultimately exacerbating malnutrition and systemic inflammation [4,10,46]. This cycle is particularly detrimental in patients with cardiovascular comorbidities, as it accelerates nutritional depletion and disease progression [4]. Targeted management, including pancreatic enzyme replacement therapy (PERT) and SIBO treatment, is therefore essential to improving long-term outcomes in these patients [10].

### 4.5. Resection Due to CD

Patients with CD already have an increased risk of SIBO due to chronic intestinal inflammation, impaired mucosal integrity, and frequent need for surgery. Resections exceeding 100 cm, ileocecal valve removal, strictures, and fibrostenosing disease have all been associated with higher SIBO prevalence [4,47]. In our cohort, SIBO prevalence after CD resections was 19%, comparable to the 31.8% reported in resected CD patients in a meta-analysis by Shah et al. [13]. Shorter resections (<100 cm) may allow for postoperative intestinal adaptation, possibly explaining the lower prevalence in our study.

In our study, longer GI resections were significantly associated with a higher risk of SIBO (*p* = 0.013), highlighting the importance of preserving bowel length whenever possible. This finding aligns with Bastos et al., who demonstrated in animal models that resection length alone, regardless of blind loop formation, predisposes to persistent SIBO [9]. While the intestine can partially adapt after moderate resections, extensive resections may exceed these compensatory mechanisms, leading not only to malabsorption but also to long-term microbial dysbiosis and metabolic complications [9,48]. Previous studies have further suggested that multiple surgeries, ileocecal valve removal, fibrostenosing disease, or combined small and large bowel resections markedly increase SIBO risk [49]. Although our sample size was limited, these findings underscore the clinical relevance of bowel length preservation and the need for larger, stratified studies to better define critical thresholds for SIBO development and its postoperative consequences.

This study has several limitations. First, the inclusion of diverse surgical procedures introduces heterogeneity, which may limit the generalisability of our findings. However, this heterogeneity reflects real-world clinical practice, where postoperative patients differ considerably in terms of anatomical changes, underlying pathology, and comorbidities. To address this, we performed stratified analyses by surgical subgroup and interpreted our results with caution, emphasising the exploratory nature of our study.

Second, no data on preoperative SIBO prevalence were available, preventing us from definitively attributing postoperative SIBO to surgery rather than to underlying disease. In addition, although we used a standardised glucose BT protocol, diagnostic limitations remain. The use of 25 g glucose instead of 75 g may have led to underestimation of SIBO prevalence; however, this choice minimised the risk of false positives from rapid glucose transit and reduced the likelihood of dumping syndrome. Similarly, while breath testing is widely used and less invasive than jejunal aspiration, the latter remains the gold standard, and diagnostic standardisation across studies is still lacking.

Third, although a post hoc power analysis confirmed adequate statistical power (>80%) for detecting moderate-to-large differences in SIBO prevalence across surgical subgroups, the relatively small sample sizes in some groups (e.g., gastric and pancreatic cancer) may have limited subgroup analyses and the detection of smaller effects.

Fourth, variability in postoperative follow-up intervals and lack of longitudinal data limit causal inference and preclude assessment of SIBO development or resolution over time. Potential confounders—including antibiotic or proton pump inhibitor use, motility disorders, dietary habits, and other lifestyle factors—were not systematically controlled, although their role in SIBO pathogenesis is well documented.

Future prospective studies using standardised diagnostic protocols, detailed nutritional and metabolic assessments, and repeated postoperative evaluations are warranted to clarify the temporal and causal relationships between surgery, SIBO, and clinical outcomes.

## 5. Conclusions

This study demonstrated a high prevalence of SIBO after extensive upper GI surgeries, with the highest rates observed following bariatric procedures, followed by gastric cancer, pancreatic, and Crohn’s disease resections. Longer resection length, adjuvant chemotherapy or radiotherapy, and certain comorbidities were significantly associated with increased SIBO risk, while clinical symptoms alone showed limited diagnostic accuracy. These findings highlight the need for systematic SIBO screening in high-risk postoperative patients. Future studies should adopt standardised diagnostic protocols and investigate whether targeted SIBO treatment improves nutritional and metabolic outcomes.

## Figures and Tables

**Figure 1 medicina-61-01822-f001:**
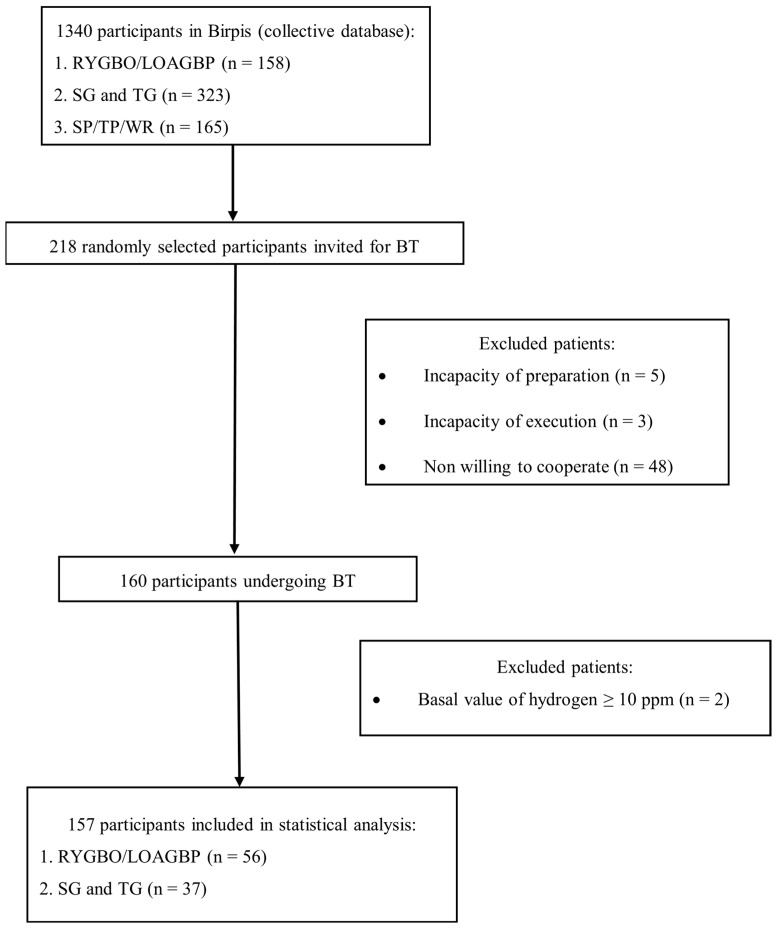
Flowchart of participants included in the final statistical analysis. BT—breath test; CD—Crohn’s disease; LOAGBP—laparoscopic one-anastomosis gastric bypass; n—number of patients; RYGBP—Roux-en-Y gastric bypass; SG—subtotal gastrectomy; SP—subtotal pancreatectomy; TG—total gastrectomy; TP—total pancreatectomy; WR—cephalic duodenopancreatectomy (Whipple’s procedure).

**Figure 2 medicina-61-01822-f002:**
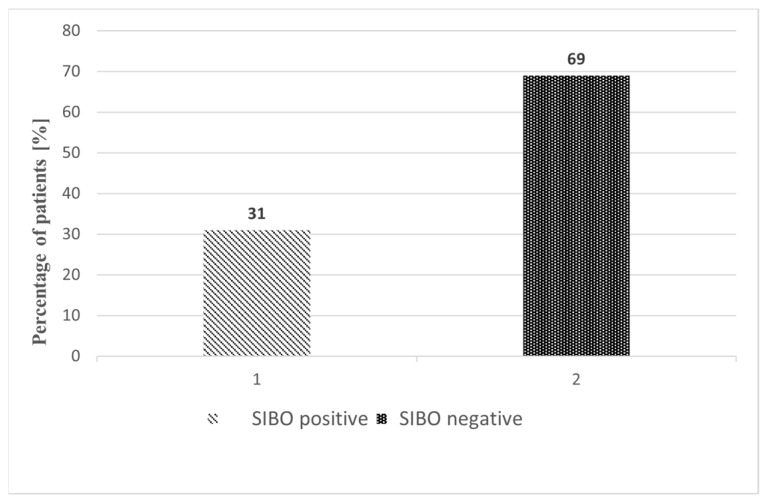
Comparison of SIBO-positive and SIBO-negative patients after upper gastrointestinal resections. Variables are expressed as percentages (%).

**Figure 3 medicina-61-01822-f003:**
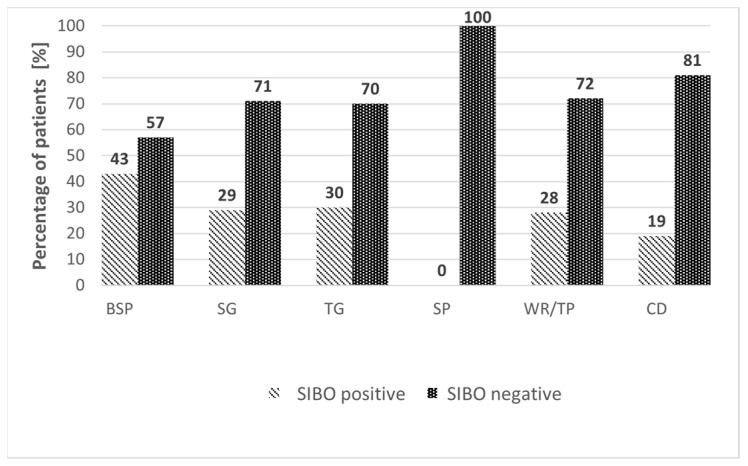
Comparison of SIBO-positive and SIBO-negative patients according to surgical subgroups of upper gastrointestinal resections. Variables are expressed as percentages (%). BSP—bariatric surgery procedures (RYGBP and LOAGBP); CD—resection of the small intestine due to Crohn’s disease; SG—subtotal gastrectomy; SP—subtotal pancreatectomy; TG—total gastrectomy; TP—total pancreatectomy; WR—cephalic pancreatectomy.

**Figure 4 medicina-61-01822-f004:**
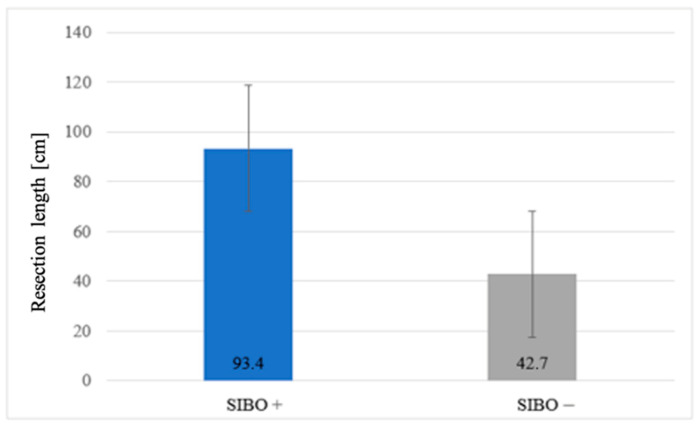
Association between resection length and SIBO prevalence.

**Table 1 medicina-61-01822-t001:** Number of patients included in the study according to the type of surgical resection.

Group	Number of Participants	Percentage (%)	Type of Surgical Resection
1	*n* = 56	36.0	RYGBP/LOAGBP
2	*n* = 37	24.0	TG/SG
3	*n* = 38	24.0	WR/TP/SP
4	*n* = 26	16.0	Resection of the small intestine due to CD
Together	*n* = 157	100	All together

Values are presented as *n* (%). SIBO—small intestinal bacterial overgrowth; RYGBP—Roux-en-Y gastric bypass; LOAGBP—laparoscopic one-anastomosis gastric bypass; SG—subtotal gastrectomy; TG—total gastrectomy; SP—subtotal pancreatectomy; TP—total pancreatectomy; WR—cephalic duodenopancreatectomy (Whipple’s procedure); CD—Crohn’s disease.

**Table 2 medicina-61-01822-t002:** Demographic characteristics of participants according to positive and negative breath tests in all participants and according to specified subgroup of surgical procedures.

**Variables**	**Positive Test (*n* = 48)**	**Negative Test (*n* = 109)**	** *p* **
All surgical procedures			
Age (years)	55.5 ± 12.9	56.9 ± 12.3	0.535
Women (n, (%))	31 (64.6)	57 (52.3)	0.153
BMI (kg/m^2^)	27.2 ± 6.1	26.4 ± 5.7	0.436
Level of education	4.52 ± 1.46	4.48 ± 1.30	0.851
SE status	5.91 ± 1.71	5.97 ± 1.70	0.848
Time from S (months)	26.1 ± 20.7	25.5 ± 16.9	0.870
**Bariatric surgery procedures**	**Positive test (*n* = 24)**	**Negative test (*n* = 32)**	** *p* **
Age (years)	49.6 ± 10.8	49.5 ± 9.5	0.842
Women (n, (%))	19 (79.2)	25 (78.1)	1.000
BMI (kg/m^2^)	30.5 ± 6.3	31.2 ± 6.8	0.673
Level of education	4.92 ± 1.35	4.66 ± 1.31	0.356
SE status	5.96 ± 1.43	5.74 ± 1.79	0.510
Time from S (months)	30.9 ±24.2	29.3 ± 15.8	0.530
**Gastric carcinoma resection**	**Positive test (*n* = 11)**	**Negative test (*n* = 26)**	** *p* **
Age (years)	59.2 ± 11.3	64.2 ± 11.3	0.272
Women (n, (%))	5 (45.5)	8 (30.8)	0.465
BMI (kg/m^2^)	24.0 ± 4.3	23.4 ± 3.4	0.654
Level of education	3.91 ± 1.38	4.15 ± 1.35	0.523
SE status	5.91 ± 1.97	6.08 ± 1.92	0.851
Time from S (months)	15.8 ± 14.9	28.8 ± 20.7	0.056
**Pancreatic carcinoma resection**	**Positive test (*n* = 8)**	**Negative test (*n* = 30)**	** *p* **
Age (years)	67.3 ± 9.1	62.1 ± 11.8	0.157
Women (n, (%))	5 (62.5)	15 (50.0)	0.697
BMI (kg/m^2^)	23.1 ± 2.9	24.3 ± 3.1	0.191
Level of education	4.13 ± 1.89	4.43 ± 1.19	0.897
SE status	5.75 ± 2.60	6.10 ± 1.54	0.499
Time from S (months)	25.6 ± 12.7	20.3 ± 15.1	0.236
**Resection due to CD**	**Positive test (*n* = 5)**	**Negative test (*n* = 21)**	** *p* **
Age (years)	57.4 ± 16.7	51.7 ± 10.0	0.297
Women (n, (%))	2 (40)	9 (42.9)	1.000
BMI (kg/m^2^)	24.8 ± 3.4	25.7 ± 4.2	0.745
Level of education	4.60 ± 1.14	4.67 ± 1.35	0.732
SE status	6.00 ± 0.82	6.00 ± 1.58	0.820
Time from S (months)	26.3 ± 20.7	23.4 ± 14.4	0.969

Values are presented as mean ± standard deviation (SD) or number (percentage). Statistical analysis was performed using the χ^2^ test, Fisher’s exact test, Mann–Whitney U test, or Student’s *t*-test, as appropriate. Statistical significance was set at *p* < 0.05. BMI—body mass index; SIBO—small intestinal bacterial overgrowth; RYGBP—Roux-en-Y gastric bypass; LOAGBP—laparoscopic one-anastomosis gastric bypass; SG—subtotal gastrectomy; TG—total gastrectomy; SP—subtotal pancreatectomy; TP—total pancreatectomy; WR—cephalic duodenopancreatectomy.

**Table 3 medicina-61-01822-t003:** Symptoms according to positive and negative breath test results in patients after upper gastrointestinal resections.

Variable (Symptom)	Positive Test (*n* = 48)	Negative Test (*n* = 109)	*p*
Chronic pain	2.38 ± 1.91	2.38 ± 1.99	0.997
Diarrhoea	3.19 ± 2.82	3.03 ± 2.51	0.724
Frequency of defecation	2.17 ± 1.00	2.04 ± 0.91	0.428
Obstipation	2.08 ± 2.22	1.82 ± 1.76	0.422
Floating stools	1.56 ± 0.50	1.71 ± 0.46	0.080
Abdominal cramps	2.64 ± 1.99	2.54 ± 2.09	0.792
Bloating and flatulence	5.85 ± 3.08	4.28 ± 2.73	0.002 *
Bloating	2.87 ± 1.23	2.50 ± 1.05	0.050 *
Nausea	1.63 ± 1.35	1.79 ± 1.71	0.557
Vomiting	1.23 ± 0.78	1.34 ± 1.17	0.552
Reflux	2.71 ± 2.38	2.70 ± 2.45	0.979
Loss of appetite	1.56 ± 1.44	1.61 ± 1.61	0.847
Bloating after meal	3.87 ± 2.79	3.40 ± 2.68	0.324
Fever	1.10 ± 0.59	1.06 ± 0.41	0.547
Joint pain	2.40 ± 1.85	2.88 ± 2.38	0.212
Fatigue	3.74 ± 2.75	3.51 ± 2.73	0.634
Memory loss or confusion	1.96 ± 1.56	2.00 ± 1.88	0.893
Mood swings	2.29 ± 1.69	2.42 ± 2.08	0.703
Belching after meals	1.67 ± 1.34	1.42 ± 1.29	0.280
Pain and bloating	1.42 ± 1.32	1.47 ± 1.24	0.799
Changing diarrhoea and obstipation	0.73 ± 1.11	0.69 ± 1.09	0.829
Diarrhoea and obstipation	0.67 ± 1.15	0.62 ± 1.01	0.776
Nausea with belching	0.63 ± 0.96	0.59 ± 0.96	0.821
Steatorrhea	0.60 ± 1.23	0.24 ± 0.71	0.711

Values are presented as mean ± standard deviation (SD). Statistical analysis was performed using the χ^2^ test, Fisher’s exact test, Mann–Whitney U test, or Student’s *t*-test, as appropriate. Statistical significance was set at *p* < 0.05. Results marked with an asterisk (*) indicate statistically significant differences between groups.

**Table 4 medicina-61-01822-t004:** Comorbidities according to positive and negative breath test results.

Comorbidity	Positive Test (*n* = 48)	Negative Test (*n* = 109)	*p*
Lactose intolerance	0.688 ± 1.36	0.312 ± 0.940	0.047 *
Systemic sclerosis	0.06 ± 0.32	0.00 ± 0.00	0.042 *
Joint pain	1.02 ± 1.31	1.07 ± 1.29	0.815
Skin problems	0.50 ± 1.19	0.49 ± 1.04	0.942
IBS	0.75 ± 1.30	0.32 ± 0.89	0.018 *
Rosacea	0.17 ± 0.63	0.06 ± 0.33	0.147
Breathing problems	0.33 ± 0.88	0.32 ± 0.83	0.933
Headache	0.71 ± 1.03	0.63 ± 0.90	0.645
Memory impairment	0.98 ± 1.23	0.82 ± 1.14	0.423
Diabetes type 1/2	1.13 ± 1.71	0.40 ± 1.15	0.002 *

Values are presented as mean ± standard deviation (SD). Statistical analysis was performed using Student’s *t*-test. IBS—irritable bowel syndrome. *p* < 0.05 indicates statistical significance. Results marked with an asterisk (*) indicate statistically significant differences between groups.

**Table 5 medicina-61-01822-t005:** (Neo)adjuvant chemotherapy and radiotherapy according to positive and negative breath test results.

(Neo)Adjuvant Therapy	Positive Test (n = 19)	Negative Test (n = 56)	*p*
CT before surgery	1.68 ± 0.478	1.79 ± 0.414	0.374
CT after surgery	1.21 ± 0.419	1.64 ± 0.484	0.001 *
RT before surgery	1.95 ± 0.229	1.95 ± 0.227	0.988
RT after surgery	1.74 ± 0.452	1.93 ± 0.260	0.027 *

This table includes only patients who underwent oncological resections of the stomach (total or subtotal gastrectomy) or pancreas (total, subtotal, or cephalic pancreaticoduodenectomy). Values are presented as mean ± standard deviation (SD). Statistical analysis was performed using the χ^2^ test, Fisher’s exact test, Mann–Whitney U test, or Student’s *t*-test, as appropriate. Statistical significance was set at *p* < 0.05. CT—chemotherapy; RT—radiotherapy. *p* < 0.05 indicates statistical significance. Results marked with an asterisk (*) indicate statistically significant differences between groups.

## Data Availability

The data presented in this study are available on reasonable request from the corresponding author. The data are not publicly available due to privacy and ethical restrictions, as they contain sensitive clinical information about individual patients. All source data are securely stored in the archives of the University Medical Centre Ljubljana (UKC Ljubljana) in accordance with institutional and national regulations.

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
