# Peer review of "Alterations in Gut Microbiota After Upper Gastrointestinal Resections: Should We Implement Screening to Prevent Complications?"

_medicina, 2025, doi:10.3390/medicina61101822_

Round 1

Reviewer 1 Report

Comments and Suggestions for Authors

the study had a very interesting idea but the prospective/ retrospective design and the very very very different surgical procedures makes very questionable conclusions

Although the modifications in the microbiota are to be expected

Author Response

For research article

Response to Reviewer 1 Comments

1. Summary

2. Questions for General Evaluation

Reviewer’s Evaluation

Response and Revisions

Does the introduction provide sufficient background and include all relevant references?

Yes

Are all the cited references relevant to the research?

Yes/Can be improved/Must be improved/Not applicable

Is the research design appropriate?

Yes/Can be improved/Must be improved/Not applicable

Are the methods adequately described?

Yes/Can be improved/Must be improved/Not applicable

Are the results clearly presented?

Yes/Can be improved/Must be improved/Not applicable

Are the conclusions supported by the results?

Yes/Can be improved/Must be improved/Not applicable

3. Point-by-point response to Comments and Suggestions for Authors

Comments 1: The study had a very interesting idea but the prospective/ retrospective design and the very very very different surgical procedures make very questionable conclusions

Although the modifications in the microbiota are to be expected.

Response 1.

Comment 1: Prospective/retrospective design.

We sincerely thank the reviewer for this important observation. We agree that the original description of the study design may have been misleading. In the revised version of the manuscript, we have clarified that this was in fact a prospective observational study, not a mixed retrospective/prospective study. The pooled surgical database (Birpis) was used only to identify patients who had undergone predefined upper gastrointestinal procedures between January 2017 and June 2022, in order to ensure adequate representation of all surgical subgroups, including rare procedures. Importantly, all clinical and diagnostic data (breath tests, questionnaires, clinical assessments) were collected prospectively between January 2021 and June 2022 according to a pre-specified protocol.

Comments 2: Different surgical procedures and questionable conclusions.

We also acknowledge that including different surgical procedures introduces heterogeneity and may limit the generalizability of conclusions. To address this, we have stratified the analyses by surgical subgroup and explicitly discussed this limitation in the Discussion section. Furthermore, we have tempered our conclusions, highlighting that our findings support the need for future studies in larger, more homogeneous cohorts, rather than providing definitive recommendations.

4. Response to Comments on the Quality of English Language

Point 1:

5. Additional clarifications

All figures, tables, the full questionnaire, and the supplementary table are included as Supplementary Material accompanying the revised manuscript, so that all data referenced in the Results section are fully accessible for review.

For review article

Response to Reviewer X Comments

1. Summary

Thank you very much for taking the time to review this manuscript. Please find the detailed responses below and the corresponding revisions/corrections highlighted/in track changes in the re-submitted files. [This is only a recommended summary. Please feel free to adjust it. We do suggest maintaining a neutral tone and thanking the reviewers for their contribution although the comments may be negative or off-target. If you disagree with the reviewer's comments please include any concerns you may have in the letter to the Academic Editor.]

2. Questions for General Evaluation

Reviewer’s Evaluation

Response and Revisions

Is the work a significant contribution to the field?

[Please give your response if necessary. Or you can also give your corresponding response in the point-by-point response letter. The same as below]

Is the work well organized and comprehensively described?

Is the work scientifically sound and not misleading?

Are there appropriate and adequate references to related and previous work? 

Is the English used correct and readable?        

3. Point-by-point response to Comments and Suggestions for Authors

Comments 1: [Paste the full reviewer comment here.]

Response 1: [Type your response here and mark your revisions in red] Thank you for pointing this out. I/We agree with this comment. Therefore, I/we have.[Explain what change you have made. Mention exactly where in the revised manuscript this change can be found – page number, paragraph, and line.]

“[updated text in the manuscript if necessary]”

Comments 2: [Paste the full reviewer comment here.]

Response 2: Agree. I/We have, accordingly, done/revised/changed/modified…..to emphasize this point. Discuss the changes made, providing the necessary explanation/clarification. Mention exactly where in the revised manuscript this change can be found – page number, paragraph, and line.]

“[updated text in the manuscript if necessary]”

4. Response to Comments on the Quality of English Language

Point 1:

Response 1:    (in red)

5. Additional clarifications

[Here, mention any other clarifications you would like to provide to the journal editor/reviewer.]

Reviewer 2 Report

Comments and Suggestions for Authors

Congratulations for the paper! an interesting read but there are some inconsistencies that I feel need to be addressed.

The methodology is unclear regarding patient selection. It is initially stated that 157 patients were identified retrospectively from 2017–2022, and that 157 patients were enrolled prospectively from 2021–2022. This gives the impression of two distinct cohorts (total n=314), which is not the case, as the results section only reports on 157 patients tested between 2021 and 2022. this needs to be clarified. are we dealing with 2 separate cohorts or the same. perhaps the authors should reconsider stating that this is both a retrospective and prospective study and that the retrospective phase only meant to identify potential patients that were meant to be included in the study. if that's not the case and there are 2 separate cohorts I believe a comparison between the 2 needs to be performed. 

The results section refers o Tables 1–9, but no tables are included in the manuscript and I could not find any additional supplements included in the submission. without these tables I cannot validated the data presented in the results section. Same goes for Figures 1-3. 

I appreciate that the authors distinctively state the limitations of their study. 

The discussion section is lengthy and I feel a bit too expansive. Maybe it could be condensed as it feels like it overshadows the main findings of the article.

the conclusion could also perhaps be made to be more concise. 

Comments on the Quality of English Language

overall the English language is good, but there are some scattered grammatical errors. the article could benefit from a bit of a polish. 

Author Response

Commentes in the word document template.

Reviewer 3 Report

Comments and Suggestions for Authors

I read with interest the manuscript entitled "Alterations in Gut Microbiota After Upper Gastrointestinal Resections: Should We Implement Screening to Prevent Complications?".

The alteration of gut microbiota following upper gastrointestinal tract resections is a significant area of research, as these surgical procedures can profoundly impact microbial composition and host health.

I suggest that the introduction be more concise. The aim at the end of the introduction is clearly stated.

Do you have an overlap in the retrospective and prospective phases? Please explain.

Remove the ethical approval from the text of the manuscript, since it is stated at the end in accordance with the template.

I can't find the tables and figures you mention in the manuscript!?

Some sentences are unclear like "An increase of ≥ ppm from baseline within 120 minutes was considered diagnostic for SIBO.".

When mentioning validated questionnaires, please provide their references to one of the previous articles in which it was used or where it was validated.

I suggest that you attach the questionnaires used as supplementary material. You must also explain in detail how the questionnaires were processed and by how many researchers?

What test did you use to test the normality of the data distribution? Furthermore, in the "statistical analysis" section, you must also state the intervals on the basis of which the correlations are interpreted.

I cannot comment on the results until the authors submit all the necessary figures and tables.

I ask that you start the discussion by presenting the most relevant results of your study, which then need to be critically interpreted in comparison with similar studies on the given topic.

Please remove a one reference where two sentences following each other use the same reference.

Much of the discussion is written in the form of a book chapter. I ask that you focus the discussion solely on comparing your results to similar studies on the topic, of which there is no shortage.

It is frivolous to cite sample size as a limitation in studies like this. Did you do a power analysis? Your database was certainly sufficient to overcome this limitation.

You are only aware of some of the limitations. Also consider and interpret the following;

  • False positives/negatives, especially if breath test protocols or interpretation criteria are inconsistent.
  • Different types of upper GI surgeries (e.g., bariatric procedures, esophagectomies, gastric surgeries) may have varying impacts on SIBO risk.
  • Variability in postoperative follow-up duration.
  • Factors like antibiotic use, proton pump inhibitor therapy, motility disorders, or other comorbidities may confound results.
  • Dietary factors and other lifestyle variables might not be adequately controlled.
  • Cross-sectional data cannot assess the development or resolution of SIBO over time post-surgery.
  •  

The conclusion is somewhat general. Please keep it concise, answering the aim of the study in just 3-4 sentences.

The references are not written according to the instructions for authors. Please correct them.

Also, there are a number of references of interest that you did not include in your manuscript. Please search the available literature in more detail.

Comments on the Quality of English Language

Please engage a native English speaker or the MDPI system to assist with English editing.

Author Response

Commentes in the word document, template bellow.

Reviewer 4 Report

Comments and Suggestions for Authors

This article evaluates the prevalence and clinical impact of small intestinal bacterial overgrowth (SIBO) after upper gastrointestinal surgery. The study investigated the incidence of SIBO and associated clinical factors in postoperative patients, analyzing 157 individuals according to the type of surgery performed. The overall incidence of SIBO was 31%, with the highest prevalence of 43% observed in patients after bariatric procedures (Roux-en-Y gastric bypass: RYGB and one-anastomosis gastric bypass: OAGB). SIBO was found to be significantly associated with symptoms such as bloating and gas, lactose intolerance, diabetes, and a history of chemotherapy or radiotherapy. Furthermore, alterations in the gut microbiota were suggested to contribute to inflammation and impaired intestinal barrier function, potentially affecting the progression of diseases such as obesity, cancer, and heart failure. The study underscores the importance of standardized diagnostic methods and early screening, indicating that appropriate treatment may contribute to better postoperative recovery and improvement in long-term quality of life. This research highlights the need for advancements in the diagnosis and management of SIBO, particularly regarding the role of gut microbiota in patients who have undergone upper gastrointestinal surgery.

However, several points of concern should be noted.

1. Crohn’s disease (CD) is already known as a condition that increases the risk of SIBO. The estimated prevalence of SIBO in patients with Crohn’s disease is approximately 20%. The underlying pathology of CD affects the structure and function of the small intestine, creating an environment prone to bacterial overgrowth. In particular, the risk of SIBO is higher when the severity of CD is greater, or when complications such as intestinal strictures and fistula formation are present, as well as when both the small and large intestines are involved. Therefore, including CD patients in the study categories may introduce considerable bias.

2. The estimated prevalence of SIBO in obese individuals is about 15%. In obesity, the balance of the gut microbiota tends to be disturbed, creating a pro-inflammatory environment. Obesity-related chronic inflammation may impair the intestinal barrier function and promote bacterial overgrowth. Additionally, reduced intestinal motility, sometimes observed in obesity, can further facilitate the proliferation of abnormal bacteria. Thus, including obesity within the investigated categories may also introduce significant bias according to the literature.

For these reasons, I think a discussion addressing the above issues is necessary.

Author Response

Comments in the word template document bellow. 

Round 2

Reviewer 2 Report

Comments and Suggestions for Authors

thank you to the authors for addressing my initial concerns. 

Author Response

Comment 1: thank you to the authors for addressing my initial concerns

Answer 1: We thank the reviewer for their kind and positive feedback.
We are pleased that the revised version of the manuscript has successfully addressed the initial concerns.
We have carefully reviewed the manuscript once again to ensure consistency and clarity before resubmission.

Reviewer 3 Report

Comments and Suggestions for Authors

Thank you for the answers and clarifications provided.

Looking at the attached tables and figures, certain discrepancies are noticeable. There are certain incorrect data. It is necessary to check all the data in detail and correct them where they are incorrect.

Furthermore, in English, a period is used with decimal numbers, not a comma. Please use an English proofreader so that basic errors are not repeated.

I also suggest that you include the most relevant tables and figures in the text of the manuscript, and not in additional materials.

Comments on the Quality of English Language

Please use the MDPI system for proofreading English, as there are still a number of basic writing errors.

Author Response

We sincerely thank the reviewer for the careful re-evaluation of our manuscript and for the helpful comments aimed at improving the quality and accuracy of the submission.
All points have been addressed in detail, as outlined below.

Comment 1: “Looking at the attached tables and figures, certain discrepancies are noticeable. There are certain incorrect data. It is necessary to check all the data in detail and correct them where they are incorrect.”

Response 1: We thank the reviewer for this valuable observation.
All tables and figures have been thoroughly re-checked against the original dataset. Minor discrepancies identified in the previous version were corrected, and all numerical data have now been verified for accuracy and consistency.
The revised tables and figures included in the manuscript reflect the corrected and validated data.

Comment 2: “Furthermore, in English, a period is used with decimal numbers, not a comma. Please use an English proofreader so that basic errors are not repeated.”

Response 2: We fully acknowledge this comment and have carefully revised the entire manuscript to ensure that all decimal numbers use periods (“.”) instead of commas (“,”), following the English formatting standard.
In addition, the manuscript underwent a comprehensive English language revision by a fluent English-speaking medical researcher and proofreading with a professional editing tool to correct minor grammatical and stylistic issues.

Comment 3: “I also suggest that you include the most relevant tables and figures in the text of the manuscript, and not in additional materials.”

Response3: We appreciate this helpful suggestion.
Following the reviewer’s advice, the most relevant tables and figures have now been integrated directly into the main manuscript text, while supplementary materials include only secondary or supporting data.
This restructuring improves the readability and scientific clarity of the paper.

Reviewer 4 Report

Comments and Suggestions for Authors

The authors have revised the manuscript extensively, and it has undergone significant improvement. 

Comments on the Quality of English Language

Modelate English revision is requierd.

Author Response

Comment 1: The authors have revised the manuscript extensively, and it has undergone significant improvement. 

Response 1: We sincerely thank the reviewer for the positive evaluation of our revised manuscript and for recognizing the significant improvement made after the previous revision.

Comment 2: Moderate English revision is required.

Response 2: We appreciate the reviewer’s suggestion regarding the language quality.
Accordingly, the entire manuscript has undergone a thorough English revision to improve grammar, clarity, and overall readability.
All changes have been implemented throughout the text, and the revised version has been carefully proofread by an English-fluent medical researcher (and further checked with a professional editing tool).